# From the betweenness centrality in street networks to structural invariants in random planar graphs

Alec Kirkley[1], Hugo Barbosa [1], Marc Barthelemy [2,3] & Gourab Ghoshal[1,4]

The betweenness centrality, a path-based global measure of flow, is a static predictor of congestion and load on networks. Here we demonstrate that its statistical distribution is invariant for planar networks, that are used to model many infrastructural and biological systems. Empirical analysis of street networks from 97 cities worldwide, along with simulations of random planar graph models, indicates the observed invariance to be a consequence of a bimodal regime consisting of an underlying tree structure for high betweenness nodes, and a low betweenness regime corresponding to loops providing local path alternatives. Furthermore, the high betweenness nodes display a non-trivial spatial clustering with increasing spatial correlation as a function of the edge-density. Our results suggest that the spatial distribution of betweenness is a more accurate discriminator than its statistics for comparing static congestion patterns and its evolution across cities as demonstrated by analyzing 200 years of street data for Paris.

[1] Department of Physics & Astronomy, University of Rochester, Rochester, NY 14627, USA. [2] Institut de Physique Théorique, CEA, CNRS-URA 2306, Gif-sur-Yvette F-91191, France. [3] Centre d'Analyse et de Mathématique Sociales (CNRS/EHESS), 54 Boulevard Raspail, Paris 75006, France. [4] Goergen Institute for Data Science, University of Rochester, Rochester, NY 14627, USA. Correspondence and requests for materials should be addressed to G.G. (email: gghoshal@pas.rochester.edu)

Recent years have witnessed unprecedented progress in our understanding of spatial networks that are pervasive in biological, technological and infrastructure systems[1,2]. These networks are quite relevant in the context of urban systems[3–7], where analysis of their structural properties has uncovered unique characteristics of individual cities, as well as surprising statistical commonalities across different urban contexts[8–10]. Patterns of streets and roads are particularly important, allowing residents to navigate the different functional components of a city. Different street structures result in varying levels of efficiency, accessibility, and usage of transportation infrastructure;[11–17] consequently structural characteristics of roads have been of great interest in the literature[18–24].

Street networks fall into the category of planar graphs[25] and their edges constitute a physical connection, as opposed to relational connections found in many complex networks[26]. The geographical embedding leads to strong effects on network topology with limitations on the number of long-range connections and the number of edges incident on a single node (its degree $k$)[27,28]. Degree-based network measures, while well-studied on such systems, lead to rather uninteresting results; the degree distribution is strongly peaked, and related metrics such as clustering and assortativity are high[2]. Instead, more information can be gleaned from non-local higher-level metrics such as those based on network centralities, which while strongly correlated with degree in non-spatial networks[29], display non-trivial behavior in planar networks[30]. Among the more studied and illuminating of such metrics is the betweenness centrality (BC), a path-based measure of the importance of a node in terms of the amount of flow passing through it[31]. More precisely, the BC for node $i$ is defined as

$$g_B(i) = \frac{1}{\mathcal{N}} \sum_{s \neq t \in V} \frac{\sigma_{st}(i)}{\sigma_{st}}, \tag{1}$$

where $\sigma_{st}$ is the number of shortest paths going from nodes $s$ to $t$ and $\sigma_{st}(i)$ is the number of these paths that go through $i$[31]. Here $\mathcal{N}$ is a normalization constant, typically of order $N^2$ where $N$ is the number of nodes, although for reasons that will be apparent later in the manuscript, we will use here the unnormalized version $\mathcal{N} = 1$.

In principle one can define a variety of different shortest paths: the number of hops in the purely topological case, the shortest distance between two points if the edges are weighted according to Euclidean distances, taking into account route preferences if edges are weighted according to a cost function such as capacity or speed-limits, or indeed some combination of the above. Incorporating this structural information into the edge-weights, the BC can be used as a proxy for predicted traffic flow[32–34]. In such a setting the paths can be considered as the optimal routes between locations, and thus nodes with high BC should expect to receive more traffic.

A number of studies have been conducted on the BC in planar graphs[35–37] finding among other things, a complicated spatial behavior of the high BC nodes[19,38], and in the case of street networks, connections to the organization and evolution of cities[39–41]. For non-planar graphs the average BC scales with the degree $k$ in a power law fashion thus $g_B(k) = \sum_{i|k_i=k} \frac{g_B(i)}{N(k)} \propto k^{\eta}$, where $N(k)$ is the number of nodes of degree $k$, and $\eta$ is an exponent depending on the graph[42]. In planar graphs, however, the BC behaves in a more complex manner, as now both topological and spatial effects are at play.

Given their practical relevance as well as the relative abundance of data, street networks have proven to be an excellent platform on which to study the properties of planar graphs including the BC. Existing analyses, however, suffer from limitations of scale

(unlike other structural properties, see ref. [43] for a recent global description), and most comparative studies of the BC across cities are typically restricted to a few square-kilometers, while studies on more extensive street-maps have been examined for at most tens of cities limited to those in Europe or North America[12,38–41]. Furthermore, there have been limited studies of the BC distribution in its entirety, with the majority of analyses instead focusing on the average BC (proportional to the average shortest path[44]) or on its maximum value[45,46].

To fill this gap in our understanding of this important class of networks, we conduct here a large-scale empirical study of the BC across 97 of the world's largest cities as measured by population (details on dataset in Methods). The cities are sampled from all six inhabited continents and the analysis is conducted at scales on the order of three thousand square-kilometers. We demonstrate that the BC distribution is an invariant quantity for most planar graphs and that it is robust to major alterations in the network, including significant changes to its topology and edge weight structure, with the relevant factors shaping the distribution being the number of nodes and edges as well as the constraint of planarity. Through simulations of random planar graph models and analytical calculations on Cayley trees, we demonstrate this to be a consequence of a bimodal regime consisting of an underlying tree structure for high BC nodes, and a low BC regime corresponding to loops providing local path alternatives. The high BC nodes display increasing spatial correlation as a function of the number of edges, leading them to cluster around the barycenter at large edge densities. The observed invariance and spatial dependence has practical implications for infrastructural and biological networks. For the case of street networks, as long as planarity is conserved, bottlenecks continue to persist, and the effect of planned interventions to alleviate structural congestion will be limited primarily to load redistribution, a feature confirmed by analyzing 200 years of data for central Paris.

## Results

**Betweenness at different scales and rescaling.** We group cities into three categories according to the number of nodes, from small ($N \sim 10^3$), medium ($N \sim 10^4$) to large road networks ($N \sim 10^5$) as shown in Fig. 1 (further details in Supplementary Note 1 and Supplementary Table 1). In Supplementary Fig. 1a, we show the betweenness probability distribution for a selection of the three categories of cities at the resolution of two and a half square-kilometers. One sees significant variability between cities, within and across categories, with mostly exponential tails (Supplementary Fig. 2) as also seen for similar samples in[39,40]. This is somewhat expected given the small sample size, and that topology of cities are different due to geographic and spatial constraints[47,48]. Indeed, variations may show up within the same city where multiple samples of a similar resolution within a city display fluctuations (Supplementary Fig. 1b). In all cases, we observe a range of behavior in the tails of the BC ranging from peaked to broad distributions, reflecting local variation in the street network structure and fluctuations in the data. One sees a dramatic difference at the scale of three thousand square-kilometers (Supplementary Figs. 1c, d) where we observe that the BC distribution for cities within each category is virtually identical, and bimodal, with two regimes separated by a bump roughly at $g_B \sim N$. For larger values of the BC we observe a slow decay signaling a broad distribution.

These trends are apparent across all 97 cities with the two regimes being separated by bumps spread across an interval of $10^3 \leq g_B \leq 10^5$ corresponding to the range of $N$ in our data (Fig. 2a). Indeed rescaling the betweenness of each node by the number of vertices in the network $g_B \to \tilde{g}_B = g_B/N$, we see the

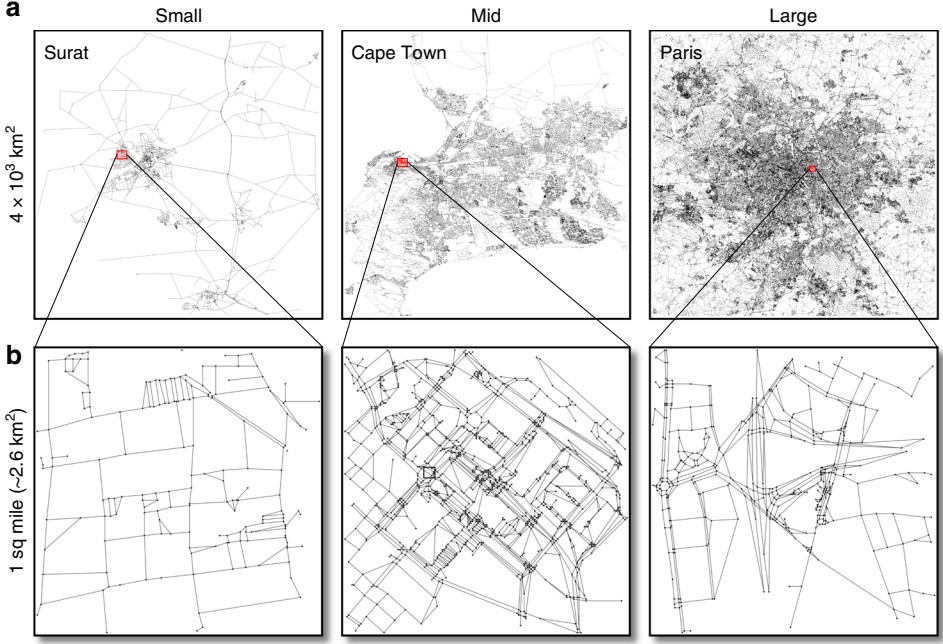

**Fig. 1** Street networks at multiple scales Cities split into three categories based on the number of nodes (intersections) in the sampled street networks: small ($N\sim10^3$), medium ($N\sim10^4$) and large ($N\sim10^5$). **a** The networks at the full sampled range. **b** Selected smaller samples (on the order of one square-mile)

distributions collapse on a single curve with a unique bump separating two clear regimes (Fig. 2b). Fitting the distribution of $\tilde{g}_B$ with the function

$$p(\tilde{g}_B) \sim \tilde{g}_B^{-\alpha} e^{-\tilde{g}_B/\beta}, \qquad (2)$$

results in a tightly bound range for $\alpha \approx 1$ and a broad size-dependent distribution for $\beta$ (Supplementary Fig. 4). Rescaling the tail with respect to $\beta$ results in a collapse of the curves for all cities (Fig. 2c). (See Supplementary Note 2, Supplementary Fig. 3 and Supplementary Table 2 for details of the rescaling and fitting procedures).

**Determinants of the betweenness centrality distribution.** Given that cities are ostensibly quite different in terms of geography or space, as well as their levels of infrastructure and socioeconomic development, the observed invariance is quite striking. To investigate the factors behind this behavior, we next systematically probe the effect of the main features that may be influencing the betweenness distribution. Examining Eq. (1), apart from its dependence on the number of nodes $N$ and the number of edges $e$, the other primary factors are the local connectivity patterns of a street intersection as governed by its degree distribution; distribution of edge weights that can correspond either to euclidean distances or some scalar quantity such as speed-limits; and planarity, the effect of space. We select the BC distribution of a number of cities as baseline and generate multiple variants of random graphs to compare with the original. In Fig. 2d we show Phoenix (blue circles) as a representative example of a city on which we perform this analysis.

To investigate the effect of varying the local neighborhood of a given street intersection, we fix the spatial position of nodes on the 2D plane and generate a Delaunay Triangulation (DT)[49] of the street network. The DT corresponds to the maximum number of edges that can be laid down between a fixed number of nodes distributed within a fixed space, without any edge-crossings. Edges are then randomly eliminated until their number

corresponds exactly to our baseline example of Phoenix. A hundred realizations of this procedure was conducted, having the effect of rewiring the local neighborhood of intersections—by changing a node's degree and its neighbors—while still maintaining planarity. In Fig. 2d we plot the average of these realizations (orange triangles), showing differences with the original street network in the lower range of the distribution, yet showing minimal change in both the location of the peak as well as the tail of the distribution. Similar random graphs were generated using a number of other cities showing the same behavior (Supplementary Fig. 5).

Next we investigate the effect of Euclidean distances on the BC distribution. We fix the number of nodes $N$ and instead of fixing their positions according to the empirical pattern, we now distribute them uniformly in the 2D plane with a scale determined by the spatial extent of the considered city. Then we generate the DT of the street network and randomly remove edges until we match the number of roads in the data. A hundred different realizations of this procedure has the effect of either dispersing high density areas or compressing very long road segments, and generating a distribution of distances that are markedly different from the original (Supplementary Fig. 8). Figure 2d (red triangles) suggests that while this has a marginally stronger effect than edge rewiring, the tails of the original and perturbed distributions are quite similar within the bounds of the error-bars. Furthermore, the positions of the peaks remain unchanged. Varying the area (and therefore density of nodes) and conducting the same procedure over multiple cities yielded identical results (Supplementary Fig. 9), suggesting that the distribution of (spatial) edge-weights has negligible effect on the BC distribution.

While the procedure outlined above does not preserve the local topology it is possible to change the edge-weights while preserving the degree sequence of nodes. This can be done by taking the original street network and randomly sampling from its associated distribution of distances, assigning each edge a number from this distribution—the edge-weights now do not

correspond to physical distances but can be interpreted instead as a cost function such as speed-limits, travel demand, or road capacity. In Fig. 2d we show the average of this process over a hundred realizations (green triangles) where each realization corresponds to a reshuffling of the edge weights over the network. While there are some changes in the distribution with a minor shift in the position of the peaks and a moderately heavier tail, no drastic modifications are apparent. Strikingly, sampling from a whole statistical family of distributions for the edge weights produced identical results (Supplementary Fig. 10), indicating little-to-no dependence on the specific nature of the weights.

Finally, we probe the effects of relaxing the constraint of planarity. Fixing $N$, the degree-sequence, and assigning weights sampled from the distance distribution of Phoenix, we use the configuration model[50] to generate one hundred non-spatial versions of the street network resulting in the markedly different curve in Fig. 2d (purple triangles). The shape of the curve is in line with the known dependence of $g_B$ on the degree for non-spatial networks, with a distribution of degrees peaked around $k = 3$ (Supplementary Figs. 11 and 12). The markedly different shape of the curve as compared to the actual street network shows that planarity appears to be the dominant factor specifying the BC distribution, with topological effects and edge-weights playing only a negligible role. While this provides an explanation for the observed similarity across cities, it does not by itself provide an explanation for the form of the distribution, its scaling with $N$, nor its bimodality, and we will provide in the following some theoretical arguments.

**Modeling the betweenness centrality distribution**. A clue for the bimodal behavior comes from the peak at $N$, a feature reminiscent of nodes adjacent to the leaves of a minimum spanning tree (MST). The MST consists of the subset of edges connecting all nodes with the minimum sum of edge-weights[51] and whose betweenness value is of $O(N)$. An examination of the BC distribution of trees therefore, may provide an explanation for the observed scaling behavior. While an exact analytical expression for the BC distribution of generalized MST's is elusive, progress can be made by approximating it as a $k$-ary tree (where each node has a branching ratio bounded by $k$). Given that the degree distribution of streets is tightly peaked (Supplementary Fig. 11), we assume a fixed branching ratio, in which case the $k$-ary tree reduces to a Cayley tree where all non-leaf nodes have degree $k$. Assuming all leaf nodes are at the same depth $L$ and adopting the convention $l = L$ for the leaf level and $l = 0$ for the root, a simple calculation reveals that for a node $v$ at level $l$, the betweenness scales as $g_B(v|k, l) \sim O(Nk^{L-l})$. After a sequence of manipulations (Methods), it can be shown that

$$P(g_B) \propto g_B^{-1}, \qquad (3)$$

indicating that the node betweenness of a Cayley tree scales with exponent $\alpha = -1$, consistent with previous calculations of the link betweenness[52]. This provides a possible explanation for the scaling with $N$ as well as the form of the tail found in the empirical measurements (Eq. (2)), indicating an underlying tree structure on which the high BC nodes of all cities lie, with the majority of flow concentrated around a spanning tree[53]. While a similar feature is seen for the BC of weighted (non-planar) random graphs, this is only true for specific families of weight distributions[54], a factor that has little-to-no effect in planar graphs.

Of course, street networks are not pure trees and contain loops given by the cyclomatic number $\Gamma = e - N + 1$ (for a connected component) where $N$ is the number of nodes and $e$ is the number of edges. In the absence of loops, $N = e + 1$, and for fixed $N$, the addition of further edges will necessarily produce loops leading to

alternate local paths for navigation. With increasing number of edges, a large fraction of the (previously) high betweenness nodes lying on the MST are bypassed, decreasing their contribution to the number of shortest paths. This induces the emergence of a low betweenness regime as well as increasingly sharp cutoffs in the tail, in line with empirical observations (Fig. 2).

To investigate the effect of increasing edges on the betweenness, we study a simple model of random planar graphs. Given that $e \sim O(N)$ and that $N$ varies over three orders of magnitude in our dataset, we define a control parameter which we call the edge density,

$$\rho_e = \frac{e}{e_{DT}}, \qquad (4)$$

defined as the fraction of extant edges $e$ compared to the maximal number of possible edges $e_{DT}$ (determined by the Delaunay triangulation). The parameter varies between $\rho_e \approx 1/3$ for the MST to $\rho_e \approx 1$ for the DT, and given that $e_{DT} \approx 3N$, this is equivalent to the ratio of edges to nodes, or in the context of street networks, the average degree $\langle k \rangle$ of street intersections[49].

Next, we distribute $N$ nodes uniformly in the 2D plane and first study the MST. To vary the density, we generate the DT on the set of nodes and remove edges until we reach the desired value for $\rho_e$. Figure 3a–d shows the betweenness distribution resulting from a hundred realizations of this procedure for $N = 10^4$ and for increasing values of $\rho_e$ from the MST to the DT. The distribution for the MST seen in Fig. 3a is peaked at $N$ and is bounded by $N^2$ which gives here a range of order $[10^4, 10^8]$. In this interval the distribution follows a form close to our calculation for the Cayley tree (Eq. (3)). As one increases $\rho_e$ and creates loops in the graph, we see the emergence of a bimodal form, with a low betweenness regime resulting from the bypassing of some of the high betweenness nodes due to the presence of alternate paths (Fig. 3b). As $\rho_e$ is further increased, the distribution gets progressively homogeneous, yet remains peaked around $N$ even as we approach the limiting case of the DT (Fig. 3d). As a guide to the eye, we shade the "tree-like" region from the "loop-like" region separated by the peak at $N$.

The simulations indicate the observed bimodality to be a combination of a high betweenness backbone belonging to the MST, and a low betweenness region generated by loops. The transition between the two regimes is determined by the minimum non-zero betweenness value for the MST, which is $O(N)$ and the tail may have different peaks, determined by the distribution of branches emanating from the tree. Progressively decorating the tree with loops leads to arbitrarily low betweenness values due to the creation of multiple alternate paths, thus smoothing out the distribution, as the betweenness transitions from an interval $[N, N^2]$ for the MST to a continuous distribution over $[1, N^2]$ for the DT.

**Spatial distribution of high betweenness centrality nodes**. Figure 3e–h shows a single instance of the actual network generated by our procedure for each corresponding edge-density. Highlighted in red are nodes lying in the 90th percentile of betweenness. There is a distinct change in spatial pattern with increasing $\rho_e$; for the MST, they span the network and are tree-like with no apparent spatial correlation; as the network gets more dense, the nodes cluster together and move closer to the barycenter, suggesting a transition between a "topological regime" and a "spatial regime".

To quantify these observed changes, we investigate the behavior of the high BC nodes at and above percentile $\theta$ through a set of metrics: the clustering $C_\theta$ which measures the spread of high betweenness nodes around their center of mass, the

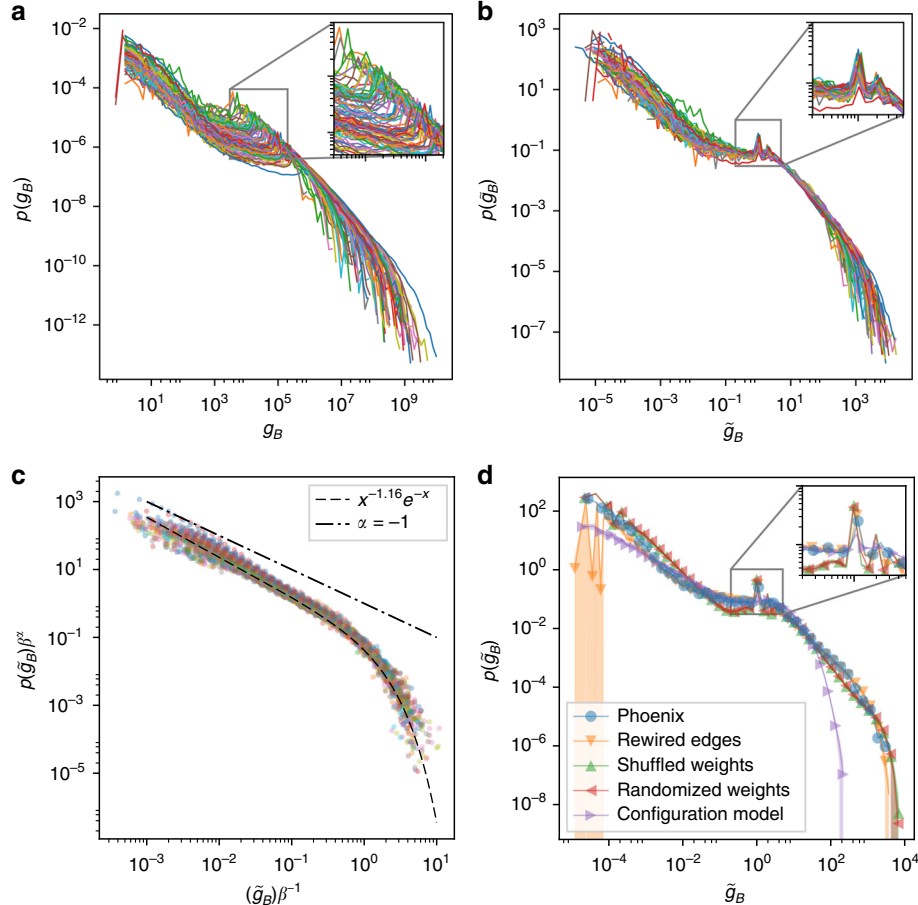

**Fig. 2** Betweenness invariance in urban streets. **a** The betweenness pdf for all 97 cities at full-resolution. The peak of the distribution for each city is shown as inset. **b** The version of the distribution after rescaling by the number of nodes $N$ showing the alignment of the peaks across all cities (also shown as inset). **c** The collapse of the tails after rescaling with respect to $\beta$. The dashed line shows the analytically computed asymptotic scaling for a Cayley-tree (Eq. (3)). **d** The BC distribution of various random graph models described in the text compared to the baseline distribution of Phoenix as a representative example. Shaded area reflects fluctuations around the average over hundred realizations of each model. Apart from the (non-spatial) configuration model, we see minimal changes in the location of the peak(s) (zoomed in inset) or shape of the tail. (Details including 2 sample KS statistics in Supplementary Figs. 6 and 7)

anisotropy factor $A_\theta$ which characterizes the spatial anisotropy of this set of nodes, and finally, the detour factor $D$ which measures the average extent to which paths between two locations deviate from their geodesic distance. (Details on metrics shown in Methods)

In Fig. 4a we plot $\langle C_\theta \rangle$ for $\theta = 90, 95$, and 97 finding a clear asymptotic decrease with increasing $\rho_e$. In Fig. 4b the plot of $\langle A_\theta \rangle$ in function of $\rho_e$, for the same set of thresholds as before, indicates a growing isotropic layout with a transition from a quasi one-dimensional to a two-dimensional spatial regime. This is confirmed by the corresponding decrease in the detour factor shown in Fig. 4c, where there is a rapid drop around $\rho_e \approx 0.4$ (or equivalently $\langle k \rangle \approx 2$) corresponding to the transition from the tree-like to the loop-like region.

Plotting the rescaled average betweenness of nodes as a function of the distance $r$ from the barycenter (Methods), demonstrates a monotonic decrease with distance in the high density regime (Fig. 4d). For low values of $\rho_e$ there appears no distance dependence of the nodes, whereas for $\rho_e > 0.4$, a clear dependence emerges with the curves converging to the form seen for maximally dense random geometric graphs as calculated in[55]. (Note that while both planar and geometric graphs are embedded in space, the latter allows for edge-crossings and therefore

broader degree distributions and larger number of edges for the same $N$. In light of this difference, the similarity between the two ostensibly different classes of graphs is notable.) In combination, the structural metrics suggest that while the spatial position of a node is decoupled from its BC value in sparse networks, a strong correlation emerges for increasingly dense networks.

We next investigate the spatial behavior of the high betweenness nodes in the empirical data. The distribution of $\rho_e$ in Fig. 5a lies in a tight range ($0.4 \leq \rho_e \leq 0.6$) with the majority of cities peaked at $\rho_e \approx 0.5$. The observed range is notable, as for one it corresponds to a range of edge densities where a clear bimodal regime exists as seen in Fig. 3, while the peaked nature of $\rho_e$ provide a further explanation for the observed similarity in BC distributions, given that it is the key controlling parameter. On the other hand, this provides a limited window for checking the spatial trends; indeed the curves for $\langle C_\theta \rangle$, $\langle A_\theta \rangle$ and $D$ shown in Fig. 5b–d are noisy. Yet, within the extent of fluctuations, the trend is reasonably consistent with that seen in Fig. 4 for the same range of $\rho_e$. A clearer picture emerges when looking at individual cities; in Fig. 5e–h we show the geospatial layout of the BC distribution for the full street network in four representative cities arranged in increasing order of $\rho_e$. Santiago, being a city with relatively sparse number of streets, shows a tree-like anisotropic

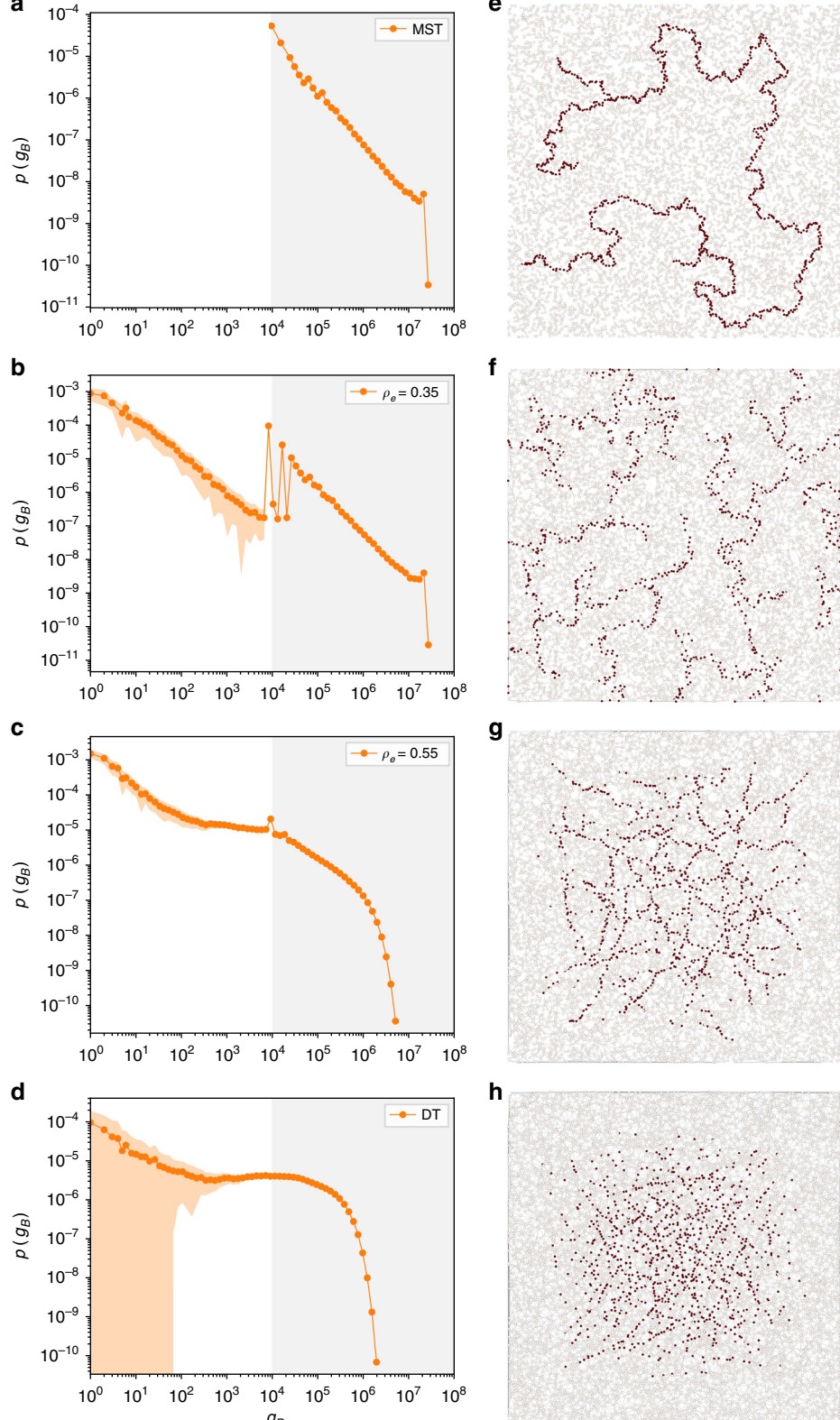

**Fig. 3** Effect of edge-density $\rho_e$ on the betweenness. $N{\sim}10^4$ nodes were randomly distributed on the 2D plane and their DT was generated. Edges were removed until the desired edge-density $\rho_e$ was reached. **a–d** The average over a hundred realizations of the resulting BC distribution ranging from the MST to the DT with increasing $\rho_e$. The orange shaded area corresponds to fluctuations around the average of the realizations, while the silver and white shades separate the "tree-like" region from the "loop-region" respectively. **e–h** a single instance of the actual generated network corresponding to each $\rho_e$. Shown in red are the nodes in the 90th percentile and above in terms of their BC value

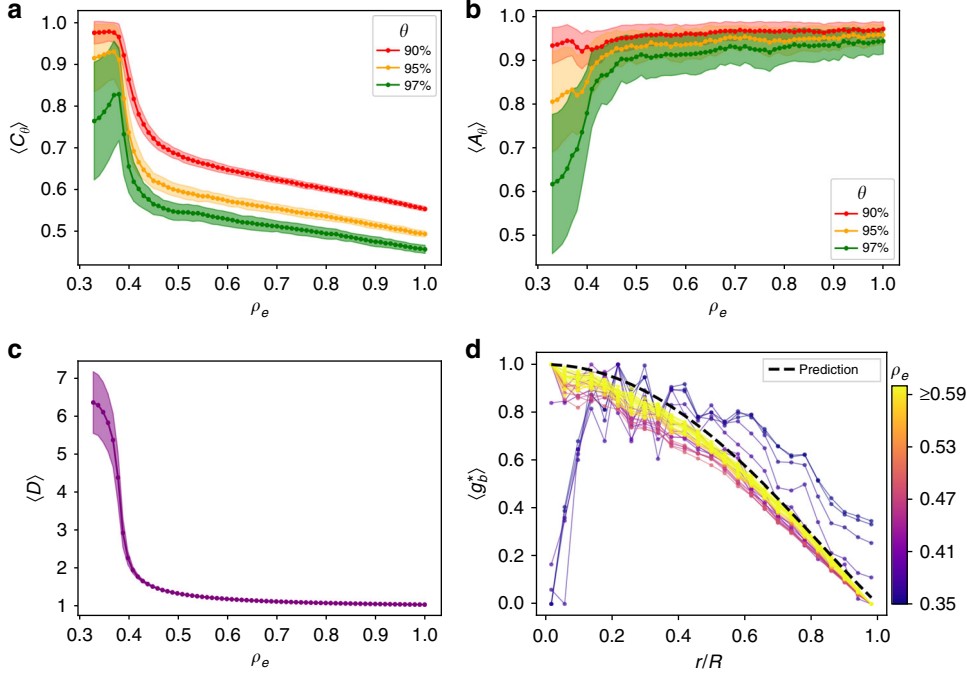

**Fig. 4** Quantifying the spatial effect of edge-density $\rho_e$ on high betweenness nodes **a** The metric $\langle C_\theta \rangle$ (Eq. (9)) decreases for denser networks, capturing the tendency of the nodes to be increasingly clustered around their center of mass. **b** Correspondingly they also become more isotropic around this center as $A_\theta$ (Eq. (11)) approaches 1. **c** The network also becomes increasingly geometric as indicated by the decrease in the average detour factor $\langle D \rangle$ (Eq. (12)) measured for the full network, which experiences an abrupt transition around $\rho_e \sim 0.4$. The shaded regions represent fluctuations over hundred realizations of the randomization procedure. **d** The average BC for nodes at a distance $r$ from the barycenter (rescaled to the interval [0,1]), measured in units of $r/R$ where $R = 50$ is the grid boundary. Curves are colored according to the value of $\rho_e$. The dashed line corresponds to the analytical calculation for an infinitely dense random geometric graph[55]. The metrics are computed for the networks generated in Fig. 3. (here $\langle \ldots \rangle$ indicates averaging over realizations)

pattern for the high BC nodes that are spread mostly along a single axis of the city. Paris and Tokyo, being in the intermediate range, show a complicated lattice-like structure with loops spanning the spatial extent of the cities. Finally, Shenyang, being a city from the upper range of densities, shows a clear (relatively symmetric) clustering of the high BC nodes around the city center.

**Temporal evolution of betweenness centrality in cities**. The changes in the structure of the random graph, shown in Fig. 3, serves as a proxy for the evolution of a city as it experiences refinements in infrastructure with increased connectivity. While historical data of complete street networks in cities is limited, progress can be made by examining smaller subsets. To this effect, we make use of five historical snapshots of a portion of central Paris spanning 200 years (1790–1999), previously gathered to study the effects of central planning by city authorities[41]. The selected portion of Paris is around thirty square kilometers with about $10^3$ intersections and road-segments, and represents the essential part of the city around 1790. This particular period was chosen to examine the effects of the so-called "Hausmann transformation", a major historical example of central planning in a city that happened in the middle of the 19th century in an effort to transform Paris and to improve traffic flow, navigability and hygiene (see refs. [41] and [56] for historical details).

In Fig. 6a we show five instances of the street network (1790, 1836, 1849, 1888, 1999), corresponding to the region clipped to 1790. Highlighted in red are nodes at and above the 90th percentile of betweenness. The spatial pattern of the nodes remains virtually identical (with a radial, spoke-like appearance) until 1849, and experiences an abrupt change to a ring-like

pattern in 1888 which persists to modern times. This change corresponds to the period after the Haussmann transformation, involving the creation of new roads, broader avenues, city squares among other things. Yet, relative to the spatial extent of the region the high betweenness nodes are located near the city center. Also of note is the relative stability of the edge-density ($\rho_e \approx 0.5$) across the temporal period, reflecting the fact that both nodes and edges are growing at the same rate (Supplementary Fig. 13).

The rescaled BC distribution, $\tilde{g}_B$, is identical for all 5 snapshots as seen in Fig. 6b despite the significant structural changes. Figure 6c, d shows the clustering $\langle C_{90} \rangle$ and anisotropy metrics $\langle A_{90} \rangle$ for the different eras, capturing the transition from the radial to the ring pattern, but are nevertheless relatively flat in correspondence with the trend in the planar random graph for fixed $\rho_e$. For purposes of comparison, we plot the averaged metrics for hundred random realizations (using the same procedure as in Fig. 3) for each of the five networks showing a remarkable similarity between the original and randomized cities. To track the evolution of the BC at the local level, we identify those intersections that are present throughout the temporal interval (within a resolution of fifty meters) and compute their betweenness in each instance of the network normalizing by $N^2$ to provide a consistent comparison, given the historical increase in intersections and roads. In Fig. 6e we plot the temporal evolution of $g_B/N^2$ for these intersections, coloring the points according to their corresponding relative rank. While one observes significant fluctuations in the BC at the local level (as expected), the high BC nodes are relatively stable from 1790 to 1849.

After the Haussmann intervention, one observes a dramatic drop in rank of the high BC nodes-corresponding to the

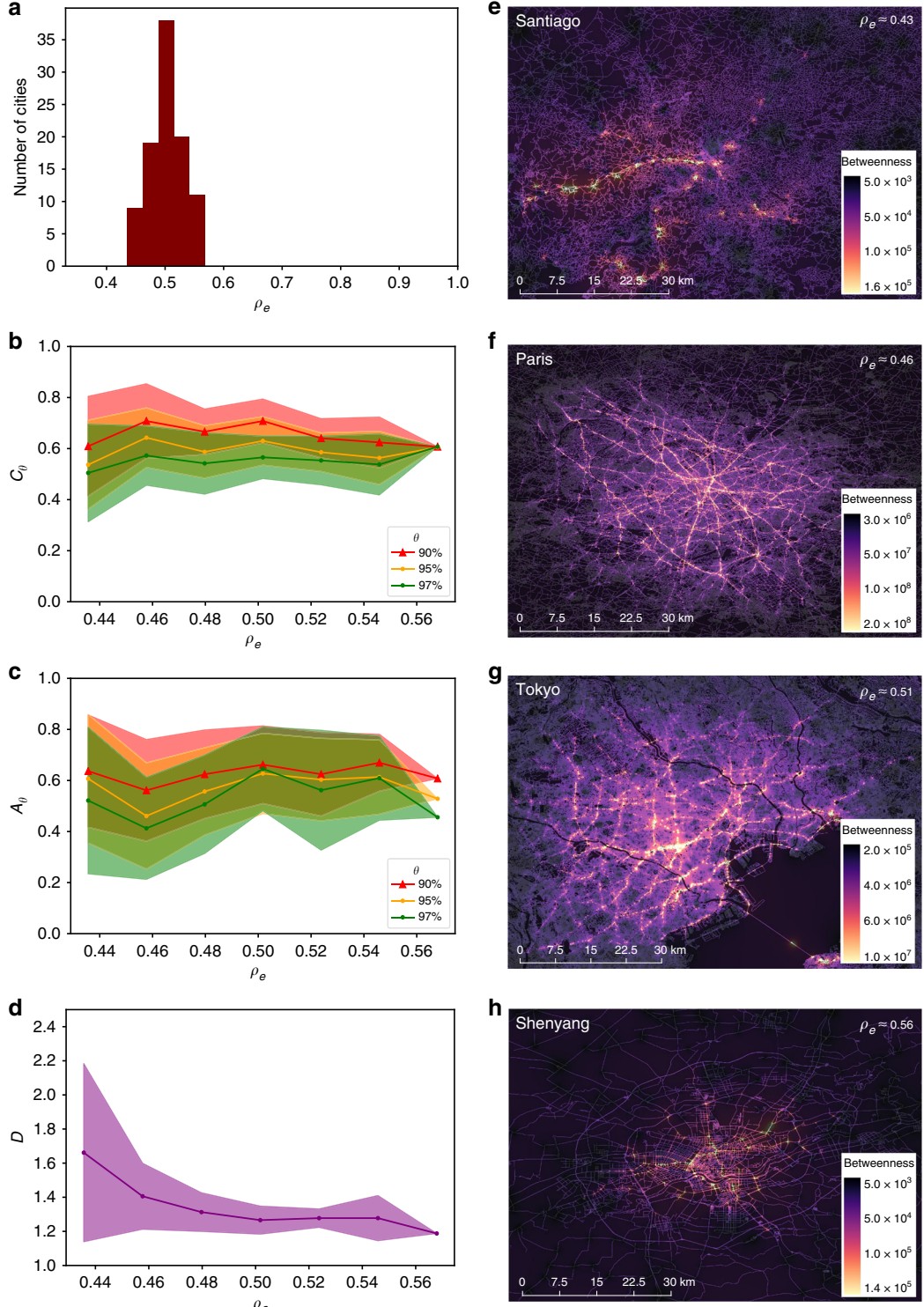

**Fig. 5** Spatial behavior of high betweenness nodes in cities. **a** Distribution of edge-densities for the 97 cities lie in a narrow range $0.4 \leq \rho_e \leq 0.6$ with most cities peaked at $\rho_e \approx 0.5$. **b** Variation of the spatial clustering $\langle C_\theta \rangle$ and **c** anisotropy ratio $\langle A_\theta \rangle$ with $\rho_e$ for the same range of thresholds used for the random graph models. **d** Detour factor for the full street network across cities plotted according to their edge-density. Points are averages over cities within a bin-size of $\rho_e = 0.02$ and the shaded areas represent the fluctuations within the bins. Fluctuations arise due to a combination of smaller samples compared to those generated in our random graph simulations, as well as the averaging over cities with the same edge-density but different $N$. **e–h** Spatial layout of intersections in four representative cities in increasing order of $\rho_e$. The color scale goes from purple to yellow with increased BC. The functional trends of the metrics and the geospatial patterns for the cities are consistent with what is observed for the random graph model described in the text

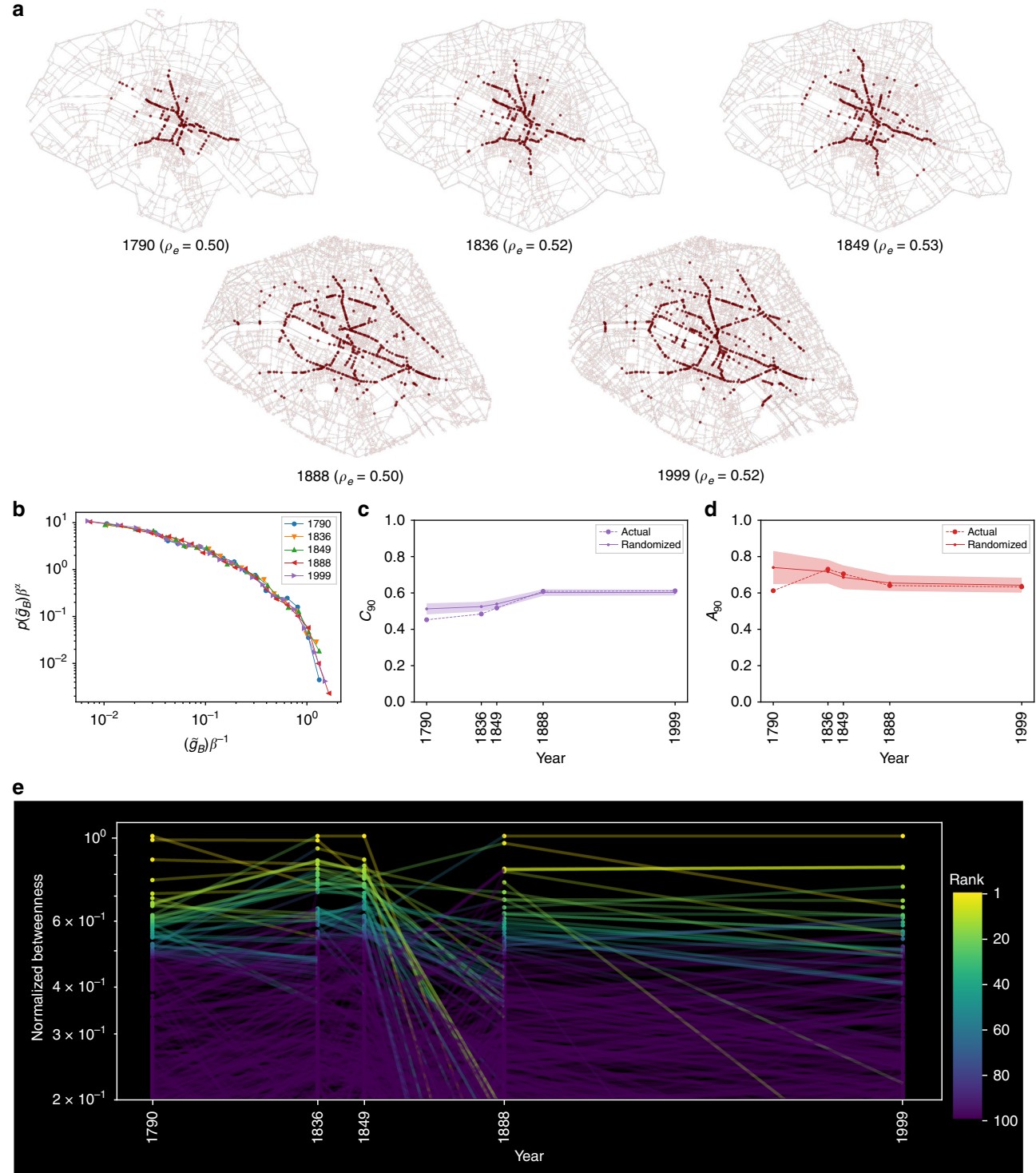

**Fig. 6** Evolution of central Paris from 1790 to 1999. **a** Five snapshots of a portion of central Paris spanning two hundred years. Colored in red are the nodes corresponding to the 90th percentile in terms of their BC. **b** The rescaled BC distributions (using the same method as in Fig. 2c) for all five networks showing that they are identical ($\alpha = 1$). **c** The clustering and **d** anisotropy metrics for the nodes above the 90th percentile. Also shown are the corresponding metrics for hundred realizations of randomized versions of the networks according to the procedure used in Fig. 3. **e** Temporal evolution of the BC of individual nodes that are present in all five networks. Points are colored according to rank based on BC and lines drawn with reduced opacity for improved visualization

"decongesting" spatial transition from a radial to a circular pattern-after which once again the high BC nodes are relatively stable till 1999. It is important to note that the load is simply redistributed to a different part of the network, as can be seen by the transition of the middle-ranked nodes to the top positions in

the same periods. Furthermore, as indicated by the spatial layout of these "new" high BC nodes, they continue to be relatively close to the center (few or none are near the periphery), a pattern that is consistent with what one would expect to find for the corresponding random graphs.

## Discussion

Taken together our results shed new light on the understanding of structural flow in spatial networks. The observed invariance in the BC distribution appears to be a function of the strong constraint imposed by planarity, leaving only the number of nodes $N$ and the number of edges $e$ as tunable parameters—a markedly different phenomenon than seen for non-planar networks, where betweenness is strongly correlated with degree. Empirical studies on street networks, analytical calculations on Cayley trees, coupled with simulations of random planar graph models, suggest this to be a consequence of a bimodal regime consisting of a tree-like structure with a tightly peaked branching ratio comprising the high betweenness "backbone" of the network, and a low betweenness regime dominated by the presence of loops. The transition of nodes between regimes is driven by increasing the density of edges in the network, which has the additional effect of introducing a spatial correlation in the high BC nodes—from being dominated by topology in the low-density regime to being strongly dependent on spatial location in the high-density regime. Given that the number of roads and intersection in our sampled cities vary over three orders of magnitude, the similarity in the BC distribution can be explained as a function of the observed narrow range of $\rho_e$. Indeed, it appears that the characteristics of flow across cities are better characterized by the spatial distribution of the high BC set, as well as the specific location of nodes that lie on this set, rather than global-level statistics.

On the other hand, the relative lack of sensitivity of the BC distribution to changes in the spatial layout, including distances and local topological variations, has interesting implications for urban planning. While the random graph models are closer in spirit to so-called self-organized cities that grow organically, the observed evolution of Paris suggests that central planning may also have its limitations. The invariance of the BC distribution suggests that congestion (in the structural sense) cannot be alleviated, but only redirected to different parts of the city. Indeed, the Haussmann transformation succeeded in doing precisely that by improving the navigability of Paris and decongesting the center. However, the high BC backbone continued to be closer to the center than the city periphery, a consequence of the spatial distribution being a function of $\rho_e$. For cities with a higher ratio of roads to intersections, the "decongestion-space" as it were, is expected to be even more limited.

It must be noted that the BC does have limitations in terms of predicting real-time traffic behavior. In particular, weighting edges based only on Euclidean distance artificially places more demand on shorter streets, although in reality, these streets may have lower speed limits and thus receive less travel demand[57]. There is also the issue of spatially irregular travel demand which is overlooked in the betweenness formulation, as all pairs of nodes are given equal weight in the calculation of the global metric[58]. Various solutions to this route-sampling issue[47] have been proposed; in particular, there have been studies using alternative versions of betweenness that weight each node pair proportional to its perceived travel demand, obtained via both real dynamic data and/or heuristics depending on the study[59,60]. The planarity constraint is also alleviated in many cases with multilevel underpasses, public transportation, etc, although the majority of the network still remains planar. We argue that despite these concerns, the results of this study are flexible enough to suggest that load redistribution will be the primary result of planned traffic intervention given static network structure. In particular, we can absorb travel preference, distance, speed limits, and other spatially heterogeneous factors into our edge weights, and the invariance of the BC distribution to edge weight adjustment can be used as evidence for these factors not affecting the global load

distribution (Cf. Supplementary Fig. 10). In addition, the construction of detours and alternative paths can be absorbed into factors affecting local topology, which also leaves the global BC distribution invariant (see Supplementary Note 3, Supplementary Fig. 14, Supplementary Table 3 for an analysis of the temporally fastest routes in a city).

Generally speaking, the study of high BC nodes is an important endeavor as they correspond to bottlenecks in networked systems. In some sense, they represent a generalization of studying the maximum BC node, that governs the behavior of the system in saturation cases where the traffic exceeds the node-capacity. Our analysis suggests, however, that for planar graphs, one needs to take into account the entire high BC set, since the maximum BC node can easily change due to local variations, yet is guaranteed to lie somewhere along the spanning tree that constitutes the backbone of the network. In this respect, further study of the mechanisms governing the spatial distribution of BC is important. Planar graphs are an important class of networks that include infrastructural systems such as power grids and communication networks, as well as transport networks found in biology and ecology[1]. In particular, leaf venation networks, arterial networks, and neural cortical networks rely on tree-like structures for optimal function[61]. The lessons from this analysis may well be gainfully employed in these other sectors.

## Methods

**Construction of street networks**. The street networks used in our analysis were constructed from the OpenStreetMaps (OSM) database[62]. For each city we extracted the geospatial data of streets connecting origin-destination pairs within a 30 km radius from the city center (referenced from https://www.latlong.net), corresponding to a rectangular area of ~60 × 60 km² with some variability due to road densities, latitude and topographical variations. The 30 km radius was chosen to encapsulate both high density urban regions and more suburban regions with fewer, longer streets. Furthermore, the choice of scale negates any (minimal) boundary effects on the calculated distribution of the BC[38,63]. The locations of the street-intersections were found using an Rtree data structure for expedited spatial search[64]. Lattitude and longitude coordinates were projected onto global distances using the Mercator projection, and adjacent intersections lying along the same roads were adjoined by edges with weights equal to the Euclidean distance between the intersections. The resulting street networks are weighted, undirected planar graphs with intersections as nodes, and edges between these nodes approximating the contour of the street network. Aggregate statistics are shown in Table 1.

**BC of Cayley trees**. Let us consider a perfect Cayley tree of size $N$ with fixed branching ratio $k$ and all leaf nodes at the same depth. Adopting the convention $l = L$ for the leaf level and $l = 0$ for the root, a node on the $l$-th level has $k-1$ branches directly below it at the $(l+1)$-th level, each with $M_{l+1}$ children such that the set of branches $\{n_i\}$ stemming from this node will have sizes $\{n_i\} = \{M_{l+1}, ..., M_{l+1}, N - M_l\}$. For fixed $k$ there are $k-1$ copies of the term $M_{l+1}$ which is of the form

$$M_\lambda = \sum_{l=0}^{L-\lambda} k^l = \frac{1 - k^{L-\lambda+1}}{1-k}. \tag{5}$$

The betweenness value of a vertex $v$ in any tree is given by $g_B(v) = \sum_{i<j} n_i n_j$ where $i, j$ are indices running over the branches coming off of $v$ (excluding $v$), and $n_i, n_j$ are the number of nodes in each branch[65]. Combining this with Eq. (5) gives us the betweenness of $v$ at level $l$ thus

$$g_B(v|k, l) = \binom{K-1}{2} M_{l+1}^2 + (k-1) M_{l+1}(N - M_l), \tag{6}$$

from which it is easy to see that for any level $l$, the betweenness scales as $g_B(v|k, l) \sim O(Nk^{L-l})$. Thus, absorbing $k^L$ into the leading constant $A$, and letting $g_B(v|k, l) \approx ANk^{-l}$, we have that since $g_B$ is completely determined by the level $l$ in which it lies in the tree,

$$P(g_B) = \sum_l P(g_B|l)P(l). \tag{7}$$

Now, using the fact that $P(l) = \frac{k^l}{N}$ and $P(g_B|l) = \delta_{g_B, ANk^{-l}}$, we have that

$$P(g_B) = A g_B^{-1}. \tag{8}$$

**Table 1 Aggregate statistics for the 97 street networks**

|          | Nodes $N$ | Edges $e$ | Length $\ell$ (km) | Area $A$ (km²) | Density $\rho$ |
|----------|-----------|-----------|--------------------|----------------|----------------|
| Mean     | 83528.87  | 130253.05 | 17461.68           | 4600.08        | 18.02          |
| St. dev  | 90335.10  | 143060.21 | 15052.83           | 1926.00        | 15.43          |
| Min      | 3349.00   | 5020.00   | 1793.45            | 777.07         | 1.00           |
| 25%      | 18925.00  | 28518.00  | 5789.36            | 3184.32        | 5.35           |
| 50%      | 62451.00  | 95797.00  | 12812.46           | 4411.81        | 14.98          |
| 75%      | 118712.00 | 178773.00 | 23751.22           | 5873.67        | 26.59          |
| Max      | 612418.00 | 976040.00 | 82586.30           | 11562.73       | 93.47          |

Shown are the average, standard deviation, minimum, maximum and various percentile values for the area $A$, number of intersections (nodes) in the network $N$, number of roads (edges) $e$, total length of streets $\ell$ and the density $\rho = N/A$ of intersections. Details for individual cities shown in Supplementary Table 1

**Spatial metrics for high BC nodes**. To measure the clustering, we specify a threshold $\theta$, i.e., we isolate nodes with a BC above the $\theta$-th percentile-and then compute their spread about their center of mass, normalizing for comparison across networks of different sizes, thus,

$$C_\theta = \frac{1}{N_\theta \langle X \rangle} \sum_{i=1}^{N_\theta} ||x_i - x_{cm}||. \quad (9)$$

Here $x_{cm} = \frac{1}{N_\theta} \sum_{i=1}^{N_\theta} x_i$, $N_\theta$ is the number of high betweenness nodes isolated, $\{x_i\}$ specify their coordinates, and $\langle X \rangle$ is the average distance of all nodes in the network to the center of mass of the high BC cluster,

$$\langle X \rangle = \frac{1}{N} \sum_{i=1}^{N} ||x_i - x_{cm}||. \quad (10)$$

Equation 9 quantifies the extent of clustering of the high BC nodes relative to the rest of the nodes in the network, with increased clustering resulting in low values of $C_\theta$.

In order to more precisely quantify the transition between the topological and spatial regimes, a clue is provided by the increasingly isotropic layout of the high BC nodes with increasing edge-density. To measure the extent of this observed (an) isotropy, we define the ratio,

$$A_\theta = \frac{\lambda_1}{\lambda_2}, \quad (11)$$

where $\lambda_1 \leq \lambda_2$ are the (positive) eigenvalues of the covariance matrix of the spatial positions of the nodes with BC above threshold $\theta$. The metric is unitless and measures the widths of the spread of points about their principal axes, analogous to the principal moments of inertia. Low values of $A_\theta$ correspond to a quasi one-dimensional structure with large anisotropy, whereas the system becomes increasingly isotropic for larger values until it is roughly two-dimensional as $A_\theta \to 1$.

The detour factor measures the average extent to which paths between two locations deviate from their geodesic distance and is given by

$$D = \frac{1}{N(N-1)} \sum_{i \neq j} \frac{d_G(i,j)}{d_E(i,j)}. \quad (12)$$

Here $d_E(i,j)$ is the euclidean distance between nodes $i, j$, and $d_G(i,j)$ is their distance-weighted shortest path in the network $G$.

**Distance dependence of BC**. In our simulations, nodes were located on a 100×100 grid with coordinates in $\mathbb{R}^2 \in [-50, 50]$. The center of the grid was chosen as the origin $(0, 0)$ and the average betweenness $\langle g_B(r) \rangle$ is computed over all nodes that are located at a distance $r$ from the origin, advancing in units of $r = 1$, until we reach the grid boundary $r = 50$. In order to restrict $\langle g_B(r) \rangle$ to the interval $[0, 1]$ we measure the rescaled quantity

$$\langle g_b^*(r) \rangle = \frac{\langle g_B(r) \rangle - \min\langle g_B(r) \rangle}{\max\langle g_B(r) \rangle - \min\langle g_B(r) \rangle}, \quad (13)$$

for different values of $\rho_e$. This was done to compare our results to the corresponding expression in random geometric graphs, which was analytically calculated for (the somewhat artificial) limit of an infinitely dense disk of radius $R$[55].

**Data availability**. All data needed to evaluate the conclusions are present in the paper and/or the Supplementary Information. The street networks were constructed from open access data. Any additional data related to this paper are available from the authors on reasonable request.

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

## Acknowledgements

This work was partially supported by the US Army Research Office under Agreement Number W911NF-17-1-0127. M.B. thanks the city of Paris (Paris 2030) for funding and the geohistoricaldata group for discussions and data. Map data copyrighted by OpenStreetMap contributors and available from https://www.openstreetmap.org.

## Author contributions

A.K., H.B., M.B., and G.G. designed the study. A.K. and H.B. implemented the method. A.K., H.B., M.B., and G.G. analyzed the results and wrote the manuscript.

## Additional information

**Competing interests:** The authors declare no competing interests.

