## [Peer Review File · Nature Communications]

Reviewers' comments:

Reviewer #1 (Remarks to the Author):

The study reports an empirical analysis of the distribution of the Betweenness Centrality (BC), one metric often used in network analysis. The authors focus on the BC distribution in a class of complex networks called planar graph. Planar graphs are graphs embedded in a two dimensional space and can be used to model a number of artificial and natural systems, thus being of great relevance across different areas of scientific investigation. The authors prove that the distribution of Betweenness Centrality in planar graphs, in particular in graphs representing urban street networks, is invariant over many different cases. They explain such invariance by means of random planar graphs simulation coupled with simple analytical calculations over 97 world cities, plus a longitudinal case on Paris in the past 200 years.

The study is rigorously conducted, well written and all the analyses have been correctly undertaken. This reviewer however detects a major problem of novelty, since the structural invariance of planar graphs and in particular the stability of BC distribution has long been thoroughly explored in a number of works in graph theory and complex networks' analysis (for example: Wang, Hernandez, and Van Mieghem, "Betweenness centrality in a weighted network". Crucitti, Porta, Latora, "Centrality Measures in Spatial Network of Urban Street").

Moreover, the dataset utilized in the manuscript are not original, as explicitly recalled by the authors (Barthelemy et al., "Self-organization versus top-down planning in the evolution of a city").

Moreover, 97 case studies of whole cities cannot be seen as a significant improvement of the state of art, which is now well into the analysis of wider samples from big data sources (for example": Domingues , Silva , Comin and da F. Costa, "Topological characterization of world cities". Jiang, "A Topological Pattern of Urban Street Networks: Universality and Peculiarity". Jiang and Jia, "Zipf's law for all the natural cities in the United States: a geospatial perspective". Barrington-Leigh and Millard-Ball, "A century of sprawl in the United States"). In this respect, a notorious issue is the way street systems are turned into graphs, whereby links commonly represent lanes, not streets, so that multi-lane roads and their intersections typically end up in a plethora of additional nodes connected by short links. This problem is not resolved in this study and may potentially have an impact on the distribution, for example contributing to generating the "bump".

As a result, in this reviewer's opinion this study, though well conducted, cannot be regarded as a novel contribution nor a significant advancement to the field. We also notice that the conclusions seem to poorly contribute to making the quantitative analysis meaningful out of the community of complex network analysts. For these reasons, the paper does not appear to meet the standard required on Nature Communication.

Reviewer #2 (Remarks to the Author):

This study demonstrates an invariant property on the distribution of betweenness centrality (BC) in street networks, in which nodes represent intersections of roads and links represent the connection of roads. The invariant property is characterized by the bimodal distribution with two regimes separated by a bump (with $BC \sim N$). The authors then use simulated planar graph models (through Delaunay Triangulation, Cayley trees et al) to investigate factors affecting the invariant and find that planarity is the dominant factor specifying the BC distribution. My comments are as follows:

1. When discussing the betweenness at different scales and rescaling, the use of scaling factor 'N' seems arbitrary. One would be keen to see the classic normalized distribution with the normalizing factor $C(n-1,2)$?
2. As the BC distributions are separated in two regimes, it is more reasonable to fit the distributions separately.
3. High BC nodes (as well as links) in reality are bounded very much to the structure of street networks which is in turn affected by both the geo-structure as well as city planning. A discussion

(or analysis) on the types of city structure and its relevance to the BC would be preferable. For example, long-chain distribution such as Santiago versus centered+loops distribution such as Shenyang.

4. Furthermore, the calculation of betweenness centrality assumes equal importance of nodes and equal amount of flows, which is far from the real flow in cities. How would the conclusions of this study be relevant to city plan (e.g., sustainable urban cities) and flow optimization is to be addressed.

5. It would be beneficial to see the comparison of other types of betweenness centralities on planar graphs presented here, e.g., the betweenness centrality based on random walk, developed in:

Newman MEJ (2005) A measure of betweenness centrality based on random walks. Social Networks 27: 39–54.

Reviewer #3 (Remarks to the Author):

The paper from A. Kirkley et al., entitled "From the betweenness centrality in street networks to structural invariants in random planar graphs" is under my opinion an outstanding piece of research in the field of spatial graphs and deserves publication. Their empirical analysis of street networks of the most populous cities worldwide sheds light to one particular pattern found for the betweenness centrality probability distribution in geographic networks, with profound implications for network flows, network dynamics and control, and obviously the recently emerging science of cities. Apart from this impressive comparative structural analysis, the paper adds another fundamental aspect in networks science: temporal evolution, usually a very difficult sphere to analyse due to data scarcity. This fact increases even more the relevance of this research, which is convincing and will influence thinking in the field.

I have though some questions that I hope authors can properly address, mostly aimed at clarifying some particular contents of the paper. I also include orthographic and typographic errors found.

Question ID Section Page Paragraph Issue

1 Introduction 4 2 "Given their practical relevance as well as (relative) abundance [...]" \diamond Why is "relative" in parenthesis?

2 Results 4 4 "In Fig. S1A we show the betweenness probability distribution" \diamond Why not the CUMULATED probability distribution? It greatly reduces histogram noise and binning problems. Moreover, later on you write "The combination of these features appear to have been overlooked or missed in existing work either due to the low resolution of the sampled street networks, or excess noise due to linear binning on a logarithmic x-axis [50]." Please, clarify this issue.

3 Results 6 Caption of Figure 2 "[...] showing the eqnarrayment of the peaks" \diamond What is this word?

4 Results 9 3 "Given that the degree distribution of cities is tightly peaked [...]" \diamond Shouldn't this "cities" be "streets"?

5 Results 11 3 "[...] as determined by its Delaunay Triangulation eDT , and which *** [...]" \diamond I guess edge density written as "rho_e" is missing here.

6 Material and methods 21 4 "For each city we extracted the geospatial data of streets connecting origin-destination pairs within a 30 kilometre radius from the city centre [...], corresponding to a rectangular area of approximately 60 x 60 km²" \diamond My most important methodological concern is related precisely with this "boundary" effect. In a planar graph (and in fact, even in a topological one), at the very moment you place a boundary that segregates your system from the rest of the universe, you determine the final (i.e., ending paths) leafs of your network. How can you be sure that patterns such as bimodality, its scaling with N or the form of the distribution are not uniquely the effect of placing a boundary in a planar graph? I believe there is no trivial hypothesis to find for this issue, since it lies at the very essence of the structural definition of a planar graph. But I would suggest including a short though or commenting about it in the discussion.

7 Supporting Information S-7 Caption Figure S2 I would like to see the error estimation of the several exponential fittings in this figure. Seeing at least an R² somewhere is a more scientific

way to convince the audience about the goodness of the fitting.

8 Supporting Information S-8 Figure S3 Are distributions (i.e., cities) randomly located in the figure? If not alphabetically, authors should clarify how this sorting has been performed.

9 Supporting Information S-22 Figure S12 Idem as Question ID #8.

10 Supporting Information S-21 Caption Figure S11 "The shaded-blue area corresponds to the standard deviation." \diamond Is it truly so? Then what is the black vertical line in each case: the average? Please clarify the caption.

Thank you very much.

Marti Rosas-Casals

Universitat Politècnica de Catalunya – BarcelonaTech

marti.rosas [at] upc.edu

Reviewer #1 (Remarks to the Author):

The study is rigorously conducted, well written and all the analyses have been correctly undertaken.

We sincerely thank the referee for this assessment and are pleased that the work is considered to be rigorously conducted, and that all analysis has been correctly undertaken.

This reviewer however detects a major problem of novelty...

We however respectfully disagree with this particular assessment. At this juncture it will be instructive to do a succinct summary of the innovative results presented in the manuscript:

- 1) The reporting of a clear bimodal regime in the global distribution of the BC, a feature that wasn't detected before and that can only be seen at the scale of this study with a large number of different road networks.
- 2) The fact that despite the significant morphological, geographic variations coupled with the design and historical evolution of the considered cities, the BC distribution is an invariant quantity after the proper rescaling.
- 3) The reasons for this invariance being explained by the constraints imposed by planarity, relegating all other considerations such as edge-weight distributions and network topology among others to a minor role as determinants, with the major role being played by the ratio of edges to nodes. **This is in great variance to what is known about the BC distribution to date, where edge weights play a disproportionately large role.**
- 4) The fact that the two distinct regimes are composed of tree-like and loop-like subsets the properties of which are thoroughly explored in our random graph model and exhaustively detailed in Figs. 3 and 4.
- 5) The reporting of the emergence of a spatial correlation between the members of the high BC set as a function of increasing edge-density (or average degree) with this set being increasingly isotropic, clustered and Euclidean as measured by a series of structural metrics in Fig. 4.
- 6) The observation of the detour factor undergoing a sudden transition, corresponding to the emergence of the spatial correlations in agreement with calculations done on an entirely separate class of graphs (geometric graph), opening up the intriguing possibility of a correspondence between two distinct classes of spatial networks (Cf. Fig 4D).
- 7) A longitudinal analysis of central Paris over 200 years, which undergoes significant change over the Haussmann period, yet all the properties related to the BC are in almost perfect agreement with our model and its predictions.

None of the 7 points listed above are known in the literature, making each of these contributions by definition novel. In combination, we contend they are **exceptionally novel**. Indeed, we have presented an unusually large number of results for a single paper, and one might argue that there is enough material here for two novel manuscripts.

“...since the structural invariance of planar graphs and in particular the stability of BC distribution has long been thoroughly explored in a number of works in graph theory and complex networks’ analysis (for example: Wang, Hernandez, and Van Mieghem, “Betweenness centrality in a weighted network”. Crucitti, Porta, Latora, “Centrality’ Measures in Spatial Network of Urban Street”).”

Yes, indeed, much has been investigated about the weighted BC distribution. We are well aware of the contributions in the literature, one of our co-authors (Barthelemy) has written a much-cited review on spatial networks (Spatial Networks, *Physics Reports* 499 (1-3), 1-101) as well as a recently published book on the subject (Morphogenesis of spatial networks, Springer, 2018). Both references the referee brought up are cited in the manuscript (Van Mieghem [55], Latora [40]), along with a host of other references (67 papers on the subject directly and indirectly relevant are cited).

We are unclear what the referee means by “structural invariance in planar graphs and in particular stability of BC distribution”. Certainly, it can’t be in the sense described in this manuscript. None of the extant papers make any references to points 1-7 listed above. In fact the two references the referee has picked **suggest the opposite of “stability” and invariance**.

Let us briefly address each of these references in turn:

Wang et al, study the distribution of weighted BC in non-spatial graphs. They are interested in what is called the overlay network, the union of shortest paths that constitute a subset of topological networks. Building upon previous work by Wu et al (Ref [54] in our manuscript) they investigate in particular the “Strong-disorder regime”. For topological graphs, it is well-known that the edge weight distribution plays a significant role in determining the BC. In this context the strong disorder regime corresponds to uniform distribution of weights (which of course is **highly unrealistic** as a model of street networks). In this regime, it is known that a disproportionately large amount of network flow is concentrated on the so-called minimal spanning tree. In the weak disorder regime (that is non-uniform weight distributions) the flow is more evenly distributed among the rest of the graph. This is not the main contribution of their paper, this feature was reported in an earlier paper (Van Mieghem and Magdalena, *Phys Rev E*, 2005), instead they do numerical simulations on networks of about 100 nodes, and report analytical calculations on a variety of tree-like structures (scale-free trees, uniform recursive trees and the k-ary tree). No comments (directly or indirectly) are made on any of the aspects listed in points 1-7 above. Indeed, their very analysis depends on the distribution of edge-weights. For planar graphs, this is not relevant. As we show the primary determinants are the number of nodes and edges. Planarity restricts all other features that are known to be important in topological networks. Note that in the planar graph, the so-called strong disorder-

der regime exists across all regimes. Indeed, a tree-like backbone exists in regimes where they do not in topological graphs. In order to uncover this backbone, one has to go through a sequence of rescaling steps that have nothing to do with the results presented in the Wang et al paper. The only correspondence between this paper and our manuscript is in the analytical calculation of the BC distribution on trees. Of course, the difference is that they calculated it for the link distribution and we did so for the node distribution. In that context we cite it correctly:

Page 9, last sentence:

While a similar feature is seen for the BC of general weighted (non-planar) random graphs, this is only true for specific families of weight distributions [55], a factor that has little-to-no-effect in planar graphs.

This correspondence constitutes a **tiny fraction** of what we report in our manuscript. The referee appears to have ignored the remaining vast majority of the paper when bringing this point up.

More crucially, there is nothing said about the invariance and stability of the distribution, nor anything about spatial correlations. In fact, the results in their paper **suggest the opposite**, as different edge-weight regimes correspond to different types of flow patterns. Even more strikingly the patterns depend on the type of tree. Van Mieghem's paper(s) is most certainly correct and constitutes a substantial contribution to the community, but it **is in the context of topological graphs**. Of course they won't see the features reported here as they were studying a **different system** and were interested in **different questions**. We therefore think that this is not comparable with our work: yes, they indeed discuss the BC but are unlike in every other conceivable manner.

The paper by Crucitti *et al* is (Ref. 40) indeed directly relevant. They do street networks across a variety of cities as planar graphs. This is not the only paper, there are two others also relevant (Refs. [12] and [41]). Indeed, one might say their work is pioneering. However, their results are in **direct contradiction** to ours. The very point of their paper is that the BC distribution is **not stable** and **not invariant** but instead **varies from city-to-city**. In particular they report two kinds of observed characteristic distributions (Gaussian and exponential) and then use this observation to characterize cities into "planned" and "self-organized". Are then their results wrong? We believe not, but they were limited by scale, which in their study is restricted to one-square-mile samples. At this scale one can reasonably come to the conclusions that they did. Indeed, this was the direct inspiration for Figure 1 in our manuscript where we show the one square mile samples in comparison to the scales we are looking at. This was also the inspiration for Figures S1A and B where we show significant variation across cities at the resolution of one square mile. More strikingly various one square mile samples taken from the same city show a similar amount of variation. In fact, this is the very point of our paper: It is misleading to use the BC distribution as a proxy for a city's organization or its evolution, since as we show in great detail that once one changes scale to the level of the entire city, it turns out the BC distribution is invariant and therefore cannot be used to make statements about a city's organization. Instead what is relevant is the spatial distribution of the high BC set, as well as the specific location of the nodes that constitute the set. We are at pains to

impress upon the referee that this is a result that is **previously unknown in the literature**. We are glad the referee brought this particular paper up, because it directly supports our claim of novelty.

Incidentally, in private communication with the corresponding author of this manuscript, one of the co-authors of the paper that the referee cited (Vito Latora) expressed a great deal of interest in our results, and is quite convinced both of its novelty and significance.

Moreover, the dataset utilized in the manuscript are not original, as explicitly recalled by the authors (Barthelemy et al., “Self-organization versus top-down planning in the evolution of a city”).

We suspect that there is some kind of misunderstanding here. We are not claiming that this dataset is novel and we explicitly refer to previous literature (as the referee correctly observes). We agree, that mere novelty of a dataset cannot be grounds for a significant contribution. What is novel, is how the data is used. In this case, a 200 years study of Paris confirms that despite the Haussmann intervention, all of our results continue to hold: the distribution of the BC is exactly the same over 200 years after rescaling, the spatial correlations match almost exactly the behavior seen in the random planar graph model, the high BC set continues to remain clustered around the center, and finally, the load is not reduced but re-distributed among the existing vertices of the street network.

We consider this to be exceptionally strong evidence of the correctness of our work.

Moreover, 97 case studies of whole cities cannot be seen as a significant improvement of the state of art, which is now well into the analysis of wider samples from big data sources (for example”: Domingues , Silva , Comin and da F. Costa, “Topological characterization of world cities”. Jiang, “A Topological Pattern of Urban Street Networks: Universality and Peculiarity”. Jiang and Jia, “Zipf’s law for all the natural cities in the United States: a geospatial perspective”. Barrington-Leigh and Millard-Ball, A century of sprawl in the United States”).

Once again, we fail to see the point here. None of the papers the referee has brought up are directly relevant to the work we present here (none of them investigate centrality), except that they too study streets at scale. (The paper by Jiang, is incidentally Ref [37] in our manuscript). There is another recent paper by Strano et al (Ref [44]) that studies the entire global street network. So while the data is not the most state-of-the-art, one can argue that it is most certainly contemporary.

In our paper, we conducted the analysis of the BC at a larger scale than prior work and led us to conclusions that contradict or refine what is known about this quantity and its distribution in planar graphs.

“In this respect, a notorious issue is the way street systems are turned into graphs, whereby links commonly represent lanes, not streets, so that multi-lane roads and their intersections typically end up in a plethora of additional nodes connected by short links. This problem is not resolved in this study and may potentially have an impact on the distribution, for example contributing to generating the “bump”.

This is not a methodological paper on how to construct street networks. We have adopted the best practices and state-of-the-art methods to construct streets. We don't see why this particular standard is being applied to us only. If this was a major issue, then all prior work on street networks come in to question. However, this problem **is** addressed by us, to the extent possible. The OSM database has information on the categories of streets, and one can (of course with some error) discard the streets that are lanes. We did so. We refer the referee to Supplementary Note S1 (Data)

The type of each street, classified into various categories by OSM ('Motorway', 'Primary', 'Service', etc.), was then added as an attribute to each edge, and two versions of the street network were created for each city. For each city, the entire street network was created, and in addition, a "refined" street network was created to approximate the network of high congestion streets, where only edges classified as primary, secondary, tertiary, highways, or service roads were kept, and all others were pruned, then the giant component of the resulting network was kept.

And

It is noteworthy that in our data, individual roads are represented as single edges, regardless of the number of traffic lanes. Roads with two or more roadways with a physical barrier separating the traffic directions (e.g., divided highways and expressways) will have one edge for each physical roadway.

The more important point is that these concerns do not affect our results one bit. The referee suggests that the over-counting of nodes and links might contribute to the bump. The suggestion here is that the bump or more accurately the bimodal regime is some kind of data artifact. If we carefully look at Figure 3 once again, we observe that over-counting the nodes and edges lead to an increased density of streets, therefore **suppressing the bump and not contributing to it.** The bimodal regime is clearly visible only at lower densities. As the network starts to fill up, it displays a transition to a continuous distribution.

The fact that the bump is so clear across the 97 cities, attests to the robustness of our results to any methodological issues in street construction. Moreover, these concerns certainly don't apply to the Paris data. Those were extracted from historical archives and hand-drawn maps. Multi-lanes were an alien concept in the 18th century. Yet our results continue to hold, rather dramatically.

Despite all of this, and to remove any doubt on the matter, we decided to conduct an additional analysis. We have added a new section to the end of the SI (Section S3) along with Fig. S14. Here we extend our analysis to actual routes sampled by residents in cities, specifically the fastest routes (based on speed-limits on streets) sourced from the Open Street map database. This constitutes a small and functional subset (see Table S3) of the full street network (taking into account planning choices and route preferences). There are no lanes on this subset. The invariance of the BC distribution along with its scaling with N continue to hold.

We have added the following excerpt as the second paragraph on page 21:

As an example in Supplementary Note S3, we show the case for route sampling. Here, we consider only those streets that lie on the temporally fastest paths (as measured by speed limits) between all Origin Destination pairs in the city, also available from the Open Street Maps (OSM) database [62]. It is important to note that these constitute a functional and comparatively much smaller (Cf. Table S3) subset of the streets and encode more information (dynamics and route sampling) than merely the spatial structure of the network. Moreover for the majority of the considered cities, the fastest paths and the shortest paths do not coincide and are markedly different [48]. In Fig. S14A we show this subset (in white) overlaid on the full street network (light red) for a city selected from each of the categories (small, medium and large). Fig. S14B shows the BC distribution for the three categories, once again indicating a clear bimodal regime with the peaks located at N , while Fig. S14C shows $\tilde{g}^b = gb/N$ with all peaks lined up. Finally, in Fig. S14D we show the rescaling of the tail with all points collapsing on to a single curve.

We also notice that the conclusions seem to poorly contribute to making the quantitative analysis meaningful out of the community of complex network analysts.

We believe that this statement is rather opaque, and we cannot respond to it unless the referee is specific. Instead we bring up the referee's earlier kind comments:

The study is rigorously conducted, well written and all the analyses have been correctly undertaken.

In summary, we have provided exceptional evidence (along with new results) that should clear any doubts about the novelty, applicability or significance of this work. We are hopeful of a positive appraisal from the referee, and humbly request an objective assessment of our manuscript.

Reviewer #2 (Remarks to the Author):

When discussing the betweenness at different scales and rescaling, the use of scaling factor 'N' seems arbitrary. One would be keen to see the classic normalized distribution with the normalizing factor $C(n-1,2)$?

It is important to note that the scaling factor N is not arbitrary. The results we present are invariant to the normalization. If one were to use the classic normalization by N^2 , the peaks will line up once again if the distribution is rescaled by multiplying it by N instead of dividing it. The choice of the un-normalized version is deliberate, as the calculation for the un-normalized version of the BC in trees scales with N (Eq. 3). Furthermore, the separation of the two regimes, the tree-like and the loop-like occur precisely at N as confirmed by the simulations of the random planar graph model as shown in Fig. 3.

As the BC distributions are separated in two regimes, it is more reasonable to fit the distributions separately.

The distributions were indeed fit separately. We fit only the tail (or the regime beyond N) since that is the regime of relevance in terms of our results. The rescaling occurs in two steps: First the N dependence is factored out, thus lining up the peaks and separating the two regimes. Then the tail is rescaled with respect to the exponential cutoff and has a truncated power-law form. Note that a pure tree is a power law exactly.

High BC nodes (as well as links) in reality are bounded very much to the structure of street networks which is in turn affected by both the geo-structure as well as city planning. A discussion (or analysis) on the types of city structure and its relevance to the BC would be preferable. For example, long-chain distribution such as Santiago versus centered+loops distribution such as Shenyang.

As we demonstrate rather conclusively in this paper, the BC distribution in a planar graph is not bounded by geo-structure or any kind of city planning. Indeed, one might say that this is the point of our contribution and why it should be considered novel. The fact that the BC distributions of 97 cities (which have significant variations in morphology, geography, different histories of evolution, different design philosophies etc) collapse onto a single curve, suggests that these factors play a limited role in the statistics of the BC, and that particular features of a given city can be seen in the spatial distribution of high BC nodes (and not in its statistics). We recognize that we should emphasize this better and were remiss in not doing so earlier, so in the abstract we added:

Furthermore, the high BC nodes display a non-trivial spatial dependence, with increasing spatial correlation as a function of the number of edges, leading them to cluster around the barycenter at large densities. Our results suggest that the spatial distribution of the BC is a more accurate discriminator when comparing patterns across cities.

And in the Discussion section (first paragraph, page 20)

Indeed, it appears that the characteristics of flow across cities is better characterized by the spatial distribution of the high BC set, as well as the specific location of nodes that lie on this set, rather than global-level statistics.

The invariance of the BC distribution was the starting point of our analysis, and is explored in great detail in the second half of the paper. Indeed, as our calculation for the tree and the simulations of the random planar graph show, the primary factors affecting planar graphs embedded in a 2D space are the number of nodes and the number of edges, or more accurately their ratio (the edge density). We understand that this might come across as a **surprising and unexpected** result for the BC on graphs since it goes counter to what is known and accepted in the literature.

It is crucial to note that prior analytical calculations of the BC on weighted graphs was conducted on non-spatial networks (where it is much easier to do so) and where topology as the referee correctly identifies, is the primary driver of the behavior. The behavior in spatial graphs is markedly different, as we exhaustively show (Figs. 2D, S5, S8, S9, S10).

In the second half of the paper we devote a large part to investigating the spatial behavior of the high BC set. As Figure 3 suggests, for low edge densities the high BC set is tree-like (as in the case for Santiago), whereas with increasing density spatial correlations emerge that not only lead to the centered+loop structure in Shenyang, but also introduces spatial dependencies on the nodes of the high BC with respect to the barycenter. They are more isotropic, more clustered, and more "Euclidean" as demonstrated by four different structural measures shown in Fig. 4.

In other words, the reason Santiago has a tree-like high BC set is because it has low edge-densities, while on the other end of the spectrum Shenyang has a higher edge density and correspondingly has a more intricate structure that is more clustered and isotropic with respect to the city center. We recall that the only factor that is important here is the ratio of nodes and edges.

More strikingly, all of these features are confirmed via a 200 years longitudinal study of the city of Paris, that includes the Haussmann intervention, a rather celebrated example of urban planning. As figure 6 shows, the distribution of the BC is exactly the same over 200 years after rescaling, the spatial correlations match almost exactly the behavior seen in the random planar graph model, the high BC set continues to remain clustered around the center, and finally, the load is not reduced but redistributed among the existing vertices of the street network.

We consider this to be an exceptionally strong demonstration of the correctness and applicability of our results.

Furthermore, the calculation of betweenness centrality assumes equal importance of nodes and equal amount of flows, which is far from the real flow in cities. How would the conclusions of this study be relevant to city plan (e.g., sustainable urban cities) and flow optimization is to be addressed.

Indeed, and this was already considered by us in the manuscript. We include the relevant paragraph for the referee for quick reference:

It must be noted that the BC does have limitations in terms of predicting real time traffic behavior. In particular, weighting edges based only on Euclidean distance artificially places more demand on shorter streets, although in reality these streets may have lower speed limits and thus receive less travel demand [58]. There is also the issue of spatially irregular travel demand which is overlooked in the betweenness formulation, as all pairs of nodes are given equal weight in the calculation of the global metric [59]. Various solutions to this route-sampling issue [48] have been proposed; in particular, there have been studies using alternative versions of betweenness that weight each node pair proportional to its perceived

travel demand, obtained via both real dynamic data and/or heuristics depending on the study [60, 61]. The planarity constraint is also alleviated in many cases with multilevel underpasses, public transportation, etc, although the majority of the network still remains planar. We argue that despite these concerns, the results in this study are flexible enough to suggest that load redistribution will be the primary result of planned traffic intervention given static network structure. In particular we can absorb travel preference, distance, speed limits, and other spatially heterogeneous factors into our edge weights, and the invariance of the BC distribution to edge weight adjustment can be used as evidence for these factors not affecting the global load distribution (Cf. Fig. S10). In addition, the construction of detours and alternative paths can be absorbed into the factors affecting local topology, which also leaves the global BC distribution invariant.

In this respect Fig. S10 is important because it considers a whole family of (spatially decoupled) weight distributions that encompasses factors such as route choices. As the figure shows, these do not make a difference to the **global** distribution of flow. Local variations may certainly emerge. Indeed if we look at only one square kilometer samples (as shown in Figure S1A), there is significant variation in the BC distribution at this scale.

Nevertheless, we decided to go beyond the random graph model and make this explicit. We have now added a new section (Supplementary Note S3, Fig. S14 and Table S3) where we take into account explicitly, route sampling preferences. Specifically, we study the temporally fastest paths between Origin Destination pairs across a range of cities, that take into account speed limits of roads and route preferences in navigation. This is metadata available from the Open Street Map Database. It is important to note, that this now corresponds to specific choices made by residents in sampling the streets and therefore contains more information than merely the structure of the street network. Additionally, the fastest routes constitute a small fraction of the overall street network (Table S3). As the figures show, nothing changes. Our results continue to hold, the distribution of the BC is bimodal, it is separated at N and it can be rescaled to collapse onto a single curve.

We have added the following excerpt as the second paragraph on page 21:

As an example in Supplementary Note S3, we show the case for route sampling. Here, we consider only those streets that lie on the temporally fastest paths (as measured by speed limits) between all Origin Destination pairs in the city, also available from the Open Street Maps (OSM) database [62]. It is important to note that these constitute a functional and comparatively much smaller (Cf. Table S3) subset of the streets and encode more information (dynamics and route sampling) than merely the spatial structure of the network. Moreover for the majority of the considered cities, the fastest paths and the shortest paths do not coincide and are markedly different [48]. In Fig. S14A we show this subset (in white) overlaid on the full street network (light red) for a city selected from each of the categories (small, medium and large). Fig. S14B shows the BC distribution for the three categories, once again indicating a clear bimodal regime with the peaks located at N , while Fig. S14C shows $\tilde{g} \tilde{b} = gb/N$ with all peaks lined up. Finally, in Fig. S14D we show the rescaling of the tail with all points collapsing on to a single curve.

We thank the referee for bringing this up, and therefore prompting us to do this analysis. We think that these additional new results (in combination with our prior exhaustive analysis) make our manuscript even more compelling than before.

It would be beneficial to see the comparison of other types of betweenness centralities on planar graphs presented here, e.g., the betweenness centrality based on random walk, developed in: Newman MEJ (2005) A measure of betweenness centrality based on random walks. Social Networks 27: 39–54.

We understand the referee's motivations in making this statement. Indeed, it is worthwhile to do a comparative analysis across different structural measures. However, this paper as it is, is quite dense, we have presented an unusually large number of results (one might argue a couple of paper's worth), and we feel that any more analysis might lead to cognitive overload and distract from the main features of the existing manuscript. Perhaps this is better left for a follow-up paper.

There are, however, some methodological problems with the referee's suggestion in using the random walk betweenness centrality. Earlier the referee suggested that the problem with the BC in terms of applicability to real cities is due to the fact that all nodes are given equal importance. Actually it's only those nodes that are on the shortest path that are given equal importance. We addressed this problem by referring to the random graph results on arbitrary weight distributions as well as conducting the new analysis on sampled routes that only give importance to nodes that lie on the fastest routes (itself a function of many other features beyond topology). By contrast the random walk centrality gives all nodes (not merely those on the shortest path) equal importance. This is perhaps even less applicable to real street networks. Furthermore, there are computational issues. The BC computed here (using the Brandes algorithm) scales as $\sim O(n^2)$ which is not ideal, but still manageable. For example, conducting our analysis on Tokyo takes about 28 hours. The random walk BC has a computational complexity of $O(n^3)$ (which is also why it's not used much for large networks). A quick scaling analysis suggest that doing a similar computation for Tokyo, will take us 8 months! Paris will take about 3 months. We are uncertain in this case, whether the rewards outweigh the temporal costs.

In summary, we thank the referee for making the thoughtful comments and prompting us to make a stronger case for the applicability of our framework to real street networks. We hope that we have addressed the concerns satisfactorily and look forward to a positive appraisal.

Reviewer #3 (Remarks to the Author):

The paper from A. Kirkley et al., entitled “From the betweenness centrality in street networks to structural invariants in random planar graphs” is under my opinion an outstanding piece of research in the field of spatial graphs and deserves publication.

We sincerely thank Dr. Rosas-Casals for the kind words and are delighted by his support for the publication of this manuscript. In particular, it is much appreciated that he took the time to go through the supplementary material, something which is often overlooked in the reviewing process. We will address the concerns using the same format as his remarks (Question ID Section Page Paragraph Issue).

1 Introduction 4 2 “Given their practical relevance as well as (relative) abundance [...]” Why is “relative” in parenthesis?

We removed the parentheses, this was a typo.

2 Results 4 4 “In Fig. S1A we show the betweenness probability distribution” Why not the CUMULATED probability distribution? It greatly reduces histogram noise and binning problems. Moreover, later on you write “The combination of these features appear to have been overlooked or missed in existing work either due to the low resolution of the sampled street networks, or excess noise due to linear binning on a logarithmic x-axis [50].” Please, clarify this issue.

We agree that the CDF is in general a better measure to plot, although the key feature (the bump separating the regimes of high and low betweenness) is washed out, and the significance of the bimodal regime is less apparent. Of course, the same analysis could easily be done on the CDF rather than the PDF. In particular, however, our intention is to accentuate the tail of the distribution that contains the high BC set, which is clearer to see in the PDF. Consequently, logarithmic binning is used as a noise reduction measure.

With respect to the second point, while conducting our (fairly thorough) literature search on the subject of betweenness in street networks, we found primarily two issues that precluded authors from observing the effects we report in this manuscript. An obvious limitation was that street samples used in previous analysis were too small and not at the scale required to see the invariance and spatial correlations that we report. Indeed, fluctuations in local topology and edge weight structure lead to the observation of strikingly different distributions for BC in the same city, as can be seen in Fig. S1(A-C). The second is simply that there is excess noise introduced in the betweenness PDF due to linear binning and the bimodal nature of the distribution is also once again washed out.

However, we recognize that these two points are rather trivial and self-evident, so we decided to remove the sentence.

“The combination of...”

3 Results 6 Caption of Figure 2 “[...] showing the eqnarrayment of the peaks” - What is this word?

Many thanks for pointing out this typo, it was supposed to say “alignment”. This has been corrected.

4 Results 9 3 “Given that the degree distribution of cities is tightly peaked [...]” Shouldn’t this “cities” be “streets”?

Yes, indeed, we have changed this to streets.

*5 Results 11 3 “[...] as determined by its Delaunay Triangulation eDT , and which *** [...]” I guess edge density written as “ ρ_e ” is missing here.*

Formatting edited in manuscript.

6 Material and methods 21 4 “For each city we extracted the geospatial data of streets connecting origin-destination pairs within a 30 kilometre radius from the city centre [...], corresponding to a rectangular area of approximately 60 x 60 km²” My most important methodological concern is related precisely with this “boundary” effect. In a planar graph (and in fact, even in a topological one), at the very moment you place a boundary that segregates your system from the rest of the universe, you determine the final (i.e., ending paths) leafs of your network. How can you be sure that patterns such as bimodality, its scaling with N or the form of the distribution are not uniquely the effect of placing a boundary in a planar graph? I believe there is no trivial hypothesis to find for this issue, since it lies at the very essence of the structural definition of a planar graph. But I would suggest including a short though or commenting about it in the discussion.

This is an extremely interesting point, and, one can think about it two ways.

The first is to do with the fact that we are truncating the street network at a putative “city boundary” as it were, and this might be creating “artificial” leaf nodes. Yet, one knows that cities typically have a dense core and relatively sparse infrastructure at the periphery. Our choice of scale (3600 km²) was chosen precisely to make sure that we do not introduce artifacts due to truncating meaningful paths. Indeed, as Figure S1 shows introducing a boundary at smaller scales (1 km²) introduces strong fluctuations in the distribution of the BC. On the other hand, as one goes up in scale the BC distribution converges to the form described here and as we show is identical across all the cities considered here, despite their significant morphological and geographic variation. The fact that it is so, suggests that the scale at which we choose the boundary is robust enough to suppress fluctuations due to noise in sampling. Crucially, the

robustness of the scaling with N and the bimodal regime is confirmed by the random planar graph model described in Figure 3 which is invariant with respect to the size of the lattice. Do also note, that one of the things we checked for in our model was the effect of assigning arbitrary Euclidean distances on the edges (street segments). This has the effect of stretching and compressing the street network and correspondingly deforming the boundary in arbitrary ways. Yet, the results continue to remain unchanged.

Finally, it turns out that the effect of boundaries on the BC has been studied recently in quite some detail. Of particular note is a recent paper by Jorge Gil, *Gil, J. Street network analysis “edge effects”: Examining the sensitivity of centrality measures to boundary conditions. Environment and Planning B: Urban Analytics and City Science 44, 819–836 (2016)*. He shows that the spatial patterns of BC and its distribution is not sensitive to boundary effects when one reaches the scale of the entire city. There is also some investigation on this in *Lion, B. & Barthelemy, M. Central loops in random planar graphs. Physical Review E 95, 042310 (2017)*, where the same conclusion is reached. We include these references on page 22, under the **Construction of Street Networks** section:

The 30 kilometer radius was chosen to encapsulate both high density urban regions and more suburban regions with fewer, longer streets. Furthermore, the choice of scale negates any (minimal) boundary effects on the calculated distribution of the BC [39, 65].

There is an additional interesting point about boundaries and planar graphs, that is not directly related to this manuscript. The variant described here is of a 2D lattice embedded on an open surface (thus necessarily including some kind of a boundary). On the other hand, in principle, one can construct planar networks embedded on arbitrary 2D closed surfaces that are homeomorphic to a sphere for example. It is possible that in such a situation our results might be altered. This might be due to the fact that there is no obvious barycenter in such networks, and therefore the notion of a spatial element is lacking. On the other hand, one can define a local barycenter and such a situation certainly deserves more study.

Most real systems, however, have a well-defined spatial boundary, although perhaps not in all dimensions. For example, in networks embedded on the surface of a cylinder (or a surface homeomorphic to it), there is once again no notion of barycenter along the angular axis, but there will be a well-defined spatial center along the axial direction. In this case, we can expect strong topological dependence on betweenness for a given axial cross section, but similar variability to that we've observed in street networks as we move along the cylindrical axis of the object. Of course, street networks in cities are not homeomorphic to either spheres or cylinders, and very much have geographic boundaries. Residents cannot teleport from one end of the city to the other. Thus, for the system that we are modeling, the choice of construction of the planar graph (as well as its attendant consequences) is appropriate.

7 Supporting Information S-7 Caption Figure S2 I would like to see the error estimation of the several exponential fittings in this figure. Seeing at least an R^2 somewhere is a more scientific way to convince the audience about the goodness of the fitting.

Thank you for pointing this out, and we regret we did not use the same stringent standards for the exponential fit as we used for the fits to the truncated power-law. The figure has been updated in the SI with R^2 values.

8 Supporting Information S-8 Figure S3 Are distributions (i.e., cities) randomly located in the figure? If not alphabetically, authors should clarify how this sorting has been performed.

9 Supporting Information S-22 Figure S12 Idem as Question ID #8.

The cities are in no particular order (just how they were stored in a python dictionary), which we've clarified in the captions for the new SI.

10 Supporting Information S-21 Caption Figure S11 "The shaded-blue area corresponds to the standard deviation." Is it truly so? Then what is the black vertical line in each case: the average? Please clarify the caption.

We've edited the caption to say "The length of the black line corresponds to the standard deviation among all cities".

Please note that we have added a new section to the end of the SI (Section S3) along with Fig. S14. Here we extend our analysis to route sampling in cities, specifically the fastest routes (based on speed-limits on streets) sourced from the Open Street map database. This constitutes a narrow and functional subset of the full street network (taking into account planning choices and route preferences). The invariance of the BC distribution along with its scaling with N continue to hold (in fact the collapse is even cleaner).

We have added the following excerpt as the second paragraph on page 21:

As an example in Supplementary Note S3, we show the case for route sampling. Here, we consider only those streets that lie on the temporally fastest paths (as measured by speed limits) between all Origin Destination pairs in the city, also available from the Open Street Maps (OSM) database [62]. It is important to note that these constitute a functional and comparatively much smaller (Cf. Table S3) subset of the streets and encode more information (dynamics and route sampling) than merely the spatial structure of the network. Moreover for the majority of the considered cities, the fastest paths and the shortest paths do not coincide and are markedly different [48]. In Fig. S14A we show this subset (in white) overlaid on the full street network (light red) for a city selected from each of the categories (small, medium and large). Fig. S14B shows the BC distribution for the three categories, once again indicating a clear bimodal regime with the peaks located at N , while Fig. S14C shows $\hat{g} \hat{b} = gb/N$ with all peaks lined up. Finally, in Fig. S14D we show the rescaling of the tail with all points collapsing on to a single curve.

In summary, we hope that we have addressed the concerns raised by Dr. Rosas-Casals and we thank him once again for his thorough and very helpful review. We are grateful for his appreciation of our work.

REVIEWERS' COMMENTS:

Reviewer #2 (Remarks to the Author):

The study analyzed a large amount of city street network data. I appreciate the effort made by the authors and that it is insightful to present the normalized betweenness centrality distribution as an invariant property for graphs with planar constraints. The authors have in addition built a model which can explain this property through simulations of random planar graph models and analytical calculations on Cayley trees. The authors have addressed all my concerns.

Reviewer #3 (Remarks to the Author):

Dear authors of NCOMMS-17-31810A-Z,
after going over your answers to my comments,
I truly believe that you have correctly addressed all of them.
Thank you very much for your time and disposition.
Yours sincerely,
M.

REVIEWERS' COMMENTS:

Reviewer #2 (Remarks to the Author):

The study analyzed a large amount of city street network data. I appreciate the effort made by the authors and that it is insightful to present the normalized betweenness centrality distribution as an invariant property for graphs with planar constraints. The authors have in addition built a model which can explain this property through simulations of random planar graph models and analytical calculations on Cayley trees. The authors have addressed all my concerns.

Reviewer #3 (Remarks to the Author):

*Dear authors of NCOMMS-17-31810A-Z,
after going over your answers to my comments,
I truly believe that you have correctly addressed all of them.
Thank you very much for your time and disposition.
Yours sincerely,
M.*

We thank the reviewers for the positive appraisal and the thoughtful comments that have significantly improved the manuscript.